# Umpolung reactivity of strained C−C σ-bonds without transition-metal catalysis

Dachang Bai [1,2] ✉, Xiuli Guo[1], Xinghua Wang [3], Wenjie Xu[1], Ruoshi Cheng[1], Donghui Wei [3], Yu Lan [3] & Junbiao Chang[1] ✉

Umpolung is an old and important concept in organic chemistry, which significantly expands the chemical space and provides unique structures. While, previous research focused on carbonyls or imine derivatives, the umpolung reactivity of polarized C−C σ-bonds still needs to explore. Herein, we report an umpolung reaction of bicyclo[1.1.0]butanes (BCBs) with electron-deficient alkenes to construct the $C(sp^3)$-$C(sp^3)$ bond at the electrophilic position of C−C σ-bonds in BCBs without any transition-metal catalysis. Specifically, this transformation relies on the strain-release driven bridging σ-bonds in bicyclo[1.1.0]butanes (BCBs), which are emerged as ene components, providing an efficient and straightforward synthesis route of various functionalized cyclobutenes and conjugated dienes, respectively. The synthetic utilities of this protocol are performed by several transformations. Preliminary mechanistic studies including density functional theory (DFT) calculation support the concerted Alder-ene type process of C−C σ-bond cleavage with hydrogen transfer. This work extends the umpolung reaction to C−C σ-bonds and provides high-value structural motifs.

Umpolung reactions create new chemical space by reversing of the inherent polarity of a functional group[1–3]. This concept provides a different approach to access target products that would be difficult to obtain by classical processes. Previous reports have mainly focused on carbonyl groups[4,5], imines[6,7] or amides[8], other reactions especially to realize the polarized C−C σ-bond umpolung, have rarely been developed and would be of much more interest.

Strain-release-driven reactions including cyclobutane, azetidine and bicyclo[1.1.1]pentane moieties have emerged as economical and efficient strategies for the construction of high-value molecular scaffolds, which are present in numerous natural products and pharmaceuticals[9–17]. Since Baran's fundamental C−N bond formation through strain-release of the bridging C−C σ-bonds in bicyclo[1.1.0] butane (BCB) derivatives[18,19], BCBs have gained significant attention from the synthetic community. BCBs have been used as privileged

motifs to give functionalized cyclobutanes through nucleophilic addition[20–25], radical addition[26–30], coupling reactions with transition-metal catalysis[31–34] and others[35–39]. BCBs served as electrophiles or radical acceptors due to their inherent electrophilic reactivity, and all these reactions occur at the β-position of the bridging C−C σ-bonds. The polarity-reversal strategy for BCBs would offer new chemical space, but remains scarce (Fig. 1A)[40,41]. In 2020, Gryko group developed the first polarity-reversal strategy of bicyclo[1.1.0]butanes (BCBs) by light-driven cobalt catalysis[40]. The in situ generation of the C−Co(III) bond is crucial for the polarity-reversal radical additions, but only gave poor diastereoselectivity of 1, 3-disubstituted cyclobutanes. Very recently, Procter group developed the SmI$_2$−catalyzed radical addition of BCB ketones to electron-deficient alkenes to afford the substituted bicyclo[2.1.1]hexanes (BCHs), which are difficult to access by other approaches (Fig. 1A)[41].

[1]State Key Laboratory of Antiviral Drugs, State Key Laboratory of Antiviral Drugs, NMPA Key Laboratory for Research and Evaluation of Innovative Drug, Key Laboratory of Green Chemical Media and Reactions, Ministry of Education, Collaborative Innovation Center of Henan Province for Green Manufacturing of Fine Chemicals, School of Chemistry and Chemical Engineering, Pingyuan Laboratory, Henan Normal University, Xinxiang 453007, China. [2]State Key Laboratory of Organometallic Chemistry, Shanghai Institute of Organic Chemistry, Chinese Academy of Sciences, Shanghai 200032, China. [3]College of Chemistry and Institute of Green Catalysis, Zhengzhou University, Zhengzhou, Henan, China. ✉e-mail: baidachang@htu.edu.cn; changjunbiao@zzu.edu.cn

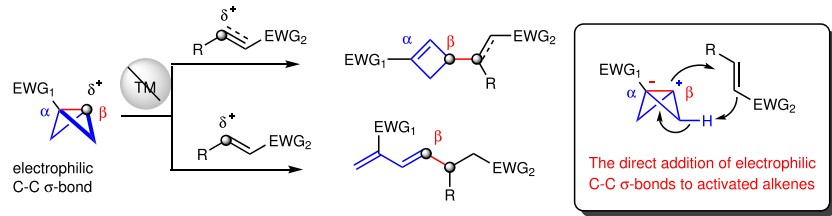

(A) Previous strategies for the transformation of bicyclo[1.1.0]butanes (BCBs)

(B) The regular Alder-ene reaction of bicyclo[1.1.0]butane (BCB) with activated alkenes

(C) Umpolung Alder-ene reaction of BCBs with electron-deficient alkenes (This work)

♦ Umpolung reaction of C–C σ-bond without transition-metal catalysis
♦ Excellent chemoselectivity for the C(sp³)-C(sp³) or C(sp²)-C(sp³) bond formation
♦ Broad functional groups tolerance with readily available substrates

**Fig. 1 | Transformation of strained C−C σ-bonds in BCBs. A** Previous strategies for the transformation of bicyclo[1.1.0]butane (BCBs). **B** The regular Alder-ene reaction of bicyclo[1.1.0]butanes (BCBs) with activated alkenes. **C** Umpolung Alder-ene reaction of BCBs with electron-deficient alkenes.

Inspired by the previous rare examples of the direct regular addition of polarized C−C σ-bonds in bicyclo[1.1.0]butane (BCBs) to activated alkenes[42–49], such as the intermolecular addition to 1,1-bis(trifluoromethyl)−2, 2-dicyanoethylene by Blanchard and Cairncross[43,44] and the intramolecular Alder-ene reaction to give spirocyclic compounds by Wipf (Fig. 1B)[48,49], we questioned whether the strain-release-driven strategy could enable umpolung reaction of the polarized C−C σ-bond with alkene. Herein, we report the umpolung reaction of BCBs with electron-deficient alkenes via Alder-ene process, delivering an array of two types of products: C(sp³)−C(sp³) bond formation products (cyclobutenes) or C(sp²)−C(sp³) bond formation products (conjugated dienes) (Fig. 1C). Based on current mechanistic studies, we propose the reaction proceeds through the concerted cleavage of C−C σ-bonds with hydrogen transfer to deliver the cyclobutene products. Moreover, the conjugated dienes could also be obtained in one pot with just modified reaction conditions.

## Results and discussion
### Reaction investigations
We initiated our studies by using bicyclo[1.1.0]butane (BCB) **1a** as model substrate for the directed addition of C−C σ-bond to activated

alkene **2a** (Fig. 2, for details see the Supplementary Table S1). After systematic investigations of the reaction parameters, we found that the polarity-reversal conjugate addition product **3a** could be obtained in 91% yield with excellent regioselectivity when dimethyl sulfoxide (DMSO) was used as the solvent at 80 °C (entry 1). Control experiments showed that the addition of Na₂SO₄, MgSO₄ and 4 Å molecular sieves could decrease the decomposition of **1a** and be benefit for the formation of **3a** (entries 2–4). Other solvents did not give a better result than DMSO (entries 5–10). The yield of **3a** was only 22% when the temperature was decreased to 50 °C, combined with starting materials recovered. The product **3a** was obtained in very low yield when the reaction was performed at 100 °C, probably due to the decomposition of **1a** at this temperature (entries 11 and 12). The concentration of this reaction was also important for efficiency (entry 13), and one equivalent of **1a** only yielded **3a** in 34% yield (entry 14). The yield of **3a** could be increased to 66% with 4.0 eq of **2a** (entry 15). Further increasing the equivalent of **2a** did not improve the yield of **3a** (for more details see the Supplementary Table S1).

### Substrates scope studies
With the optimized reaction conditions in hand, we then explored the generality of this umpolung reaction (Fig. 3). Using bicyclo[1.1.0]

| entry | Variation from the "standard" conditions | 3a, yield (%) |
|---|---|---|
| 1 | none | 91 |
| 2 | no Na$_2$SO$_4$ | 30 |
| 3 | 4Å instead of Na$_2$SO$_4$ | 38 |
| 4 | MgSO$_4$ instead of Na$_2$SO$_4$ | 82 |
| 5 | DMF instead of DMSO | 52 |
| 6 | DCE instead of DMSO | 54 |
| 7 | MeOH instead of DMSO | 54 |
| 8 | toluene instead of DMSO | 42 |
| 9 | dioxane instead of DMSO | 58 |
| 10 | THF instead of DMSO | 42 |
| 11 | 50 °C instead of 80 °C | 22 |
| 12 | 100 °C instead of 80 °C | 6 |
| 13 | 0.2 mL DMSO instead of 0.1 mL DMSO | 75 |
| 14 | 1 eq 1a instead of 2 eq 1a | 34 |
| 15$^b$ | 4 eq 2a was used | 66 |

**Fig. 2 | Optimization of reaction conditions.$^a$ a** Reaction conditions A: BCB **1a** (0.2 mmol), alkene **2a** (0.1 mmol), Na$_2$SO$_4$ (0.17 mmol), 80 °C in DMSO (0.1 mL) under argon, 24 h, isolated yield. **b** Reaction conditions B: BCB **1a** (0.1 mmol), alkene **2a** (0.4 mmol).

butane **1a** as a standard substrate, the scope of electro-deficient alkenes was examined and found to be very broad. A series of functional groups at the *para*-position of benzene rings worked well to give the desired products (**3a**–**3f**, 51–91% yield). The *meta*- and *ortho*-substituted enones are also tolerated (**3g**–**3k**, 44–68% yield). Hetero-aryl substituted enones also gave the corresponding products **3l**–**3o** in 58–85% yield. The alkenes with alkyl-substituted, ester or aldehyde group also worked smoothly to give C(sp$^3$)–C(sp$^3$) bond formation products (**3p**–**3r**, 39–72% yield). Other amide substituted BCBs were also investigated and found to be compatible with Weinreb amide (**3s**, 79% yield) and morpholine amide (**3t**, 70% yield). Besides amides, various electron-withdrawing groups such as sulfone, ester and nitrile were successfully converted to the corresponding products just with neat reaction conditions (**3u**, 12% yield; **3v**, 42% yield and **3w**, 24% yield). *Aryl* ketones and *alkyl* ketone attached BCBs also worked smoothly to give umpolung Alder-ene products (**3x**, 34% yield; **3xa**, 59% yield; **3xb**, 39% yield and **3xc**, 34% yield) (for details see the Supplementary Tables S2 and S3). Given the long synthesis route of bicyclo[1.1.0]butanes, excess alkenes were used for several substrates (Reaction conditions B), but led to lower yield of desired products (**3a**, 66% yield; **3c**, 41% yield; **3i**, 39% yield; **3j**, 65% yield; **3m**, 60% yield; **3q**, 19% yield and **3s**, 67% yield). Noteworthy, the substituted alkyne is also suitable to be an acceptor in this reaction system and a high yield was obtained with excess of alkyne (**3y**, 72% yield). However, the reaction with electron-rich alkenes such as styrene and 1,1-diphenylethylene failed to give addition products, only with starting materials recovered.

To better define the application of this coupling system, we then examined the compatibility of more challenged 1, 2-disubstituted alkenes. The β-fluoroalkyl enones are attractive building blocks for the synthesis of high-value-added organofluorine compounds[50–53]. To our delight, these fluoroalkyl-containing enones are successfully applied in our systems, affording the C(sp$^3$)–C(sp$^3$) bond formation products **4** and C(sp$^2$)–C(sp$^3$) bond formation products **5** with excellent selectivity, respectively (Fig. 4, for optimization of reaction conditions see the Supplementary Table S4). We first briefly examined the substrate

scope to deliver cyclobutenes **4**. A series of enones bearing *fluoro-*, *chloro-*, electron-withdrawing and -donating groups at the *para*-position of the benzene ring worked smoothly to give formal 1, 4-conjugate addition products with excellent yield, moderate diastereoselectivity and excellent regioselectivity (**4a**–**4f**, 74–99% yield, *dr* 5: 1, **4:5** = 8:1 to >20:1). The reaction also tolerated with 2-naphthyl-substituted enone (**4g**, 99% yield, *dr* 5: 1, **4:5** > 20:1). When the trifluoromethyl group was replaced by a pentafluoroethyl group, the desired cyclobutene **4h** was obtained in 98% yield and excellent diastereoselectivity (*dr* 10:1, **4:5** > 20:1). Other electron-withdrawing groups such as Weinreb amide (**4i**, 36% yield), ester (**4j**, 21% yield) and alkyl ketone (**4k**, 34% yield) did lead to umpolung addition product formation, albeit with low yields. Interestingly, we got the C(sp$^2$)–C(sp$^3$) bond formation diene product **5** when the reaction proceeded at higher temperature, and no cyclobutenes **4** were observed under this condition. Again, the substrate scope to access **5** was also found to be very broad. Various enones bearing *halo-*, electron-withdrawing and -donating groups were well tolerated (**5a**–**5h**, 62–95% yield). The structure of **5g** was unambiguously confirmed by X-ray crystallography (CCDC 2216789). The enones bearing *fluoro-*, *chloro-*, and trifluoromethyl groups at the *meta*-position of the benzene rings gave the desired products in moderate yield (**5i**–**5k**, 67–68% yield). The introduction of a substituent at the *ortho* position resulted in slightly diminished yield (**5l**–**5n**, 47–69% yield), likely due to the steric hindrance effect. The 2-naphthyl- and furyl-substituted enones afforded the corresponding products in acceptable yield (**5o**, 54% yield and **5p**, 44% yield). An array of functionalized β-pentafluoroethyl enones also successfully led to the diene products (**5q**–**5u**, 44–64% yield). However, the less reactive (*E*)-1-phenylbut-2-en-1-one, chalcone and bicyclo[2.1.0]pentane all failed to give conjugate addition products and dienes. These results in the Synthetic applications and Computational studies indicated diene product **5a** was formed from **4a** through electrocyclic ring-opening process.

Furthermore, we investigated the influence of the substituent at the β-position of BCBs in this umpolung reaction. As shown in Fig. 5, with the 1,3-disubstituted bicyclobutane **1a'**, some differently substituted enones and acrolein worked well to deliver the regular addition products at the α-position of the electron-withdrawing group (**6a**–**6d**, 59–99% yield), but not the polarity-reversal addition products at the β-position[54–57]. The structure of **6b** was confirmed by X-ray crystallography (CCDC 2217250). Moreover, the amide group attached to BCBs proved to be viable substrates, providing addition products in excellent yield (**6e**, 97% yield and **6f**, 99% yield). These results indicated the steric hindrance of the substituent at the β-position would change the regioselectivity of the bridging C–C σ-bond in our reaction systems.

## Synthetic applications

This polarity-reversal C–C σ-bond involved in the conjugate addition reaction turned out to be synthetically useful (Fig. 6). The reaction was readily scaled up to 6.0 mmol, giving the cyclobutene product **3r** in 70% yield and the diene product **5a** in 76% yield, respectively. The diene product **5a** could be obtained quantitatively from **4a** through an electrocyclic ring-opening process and was ready to undergo isomerization under base conditions to give product **7** in 42% yield. The cyclobutene **3r** was successfully hydrogenated by H$_2$ to give cyclobutane product **8** in 73% yield and 10:1 *dr* in the presence of Pd/C catalyst. When Raney-Ni was used, the reduction of carbonyl group and double bond occurred in one pot to give cyclobutane product **9** in 68% yield and 17:1 *dr*[58,59]. Other reactions such as Wittig-reactions were also successfully performed, giving the product **10** in 93% yield or terminal alkene **11** in 81% yield. The selective reduction of carbonyl group with NaBH$_4$ could give the alcohol product **12** in 85% yield, which would undergo cyclization under base conditions to access bicyclic product **13** in 90% yield and 6:1 *dr*.

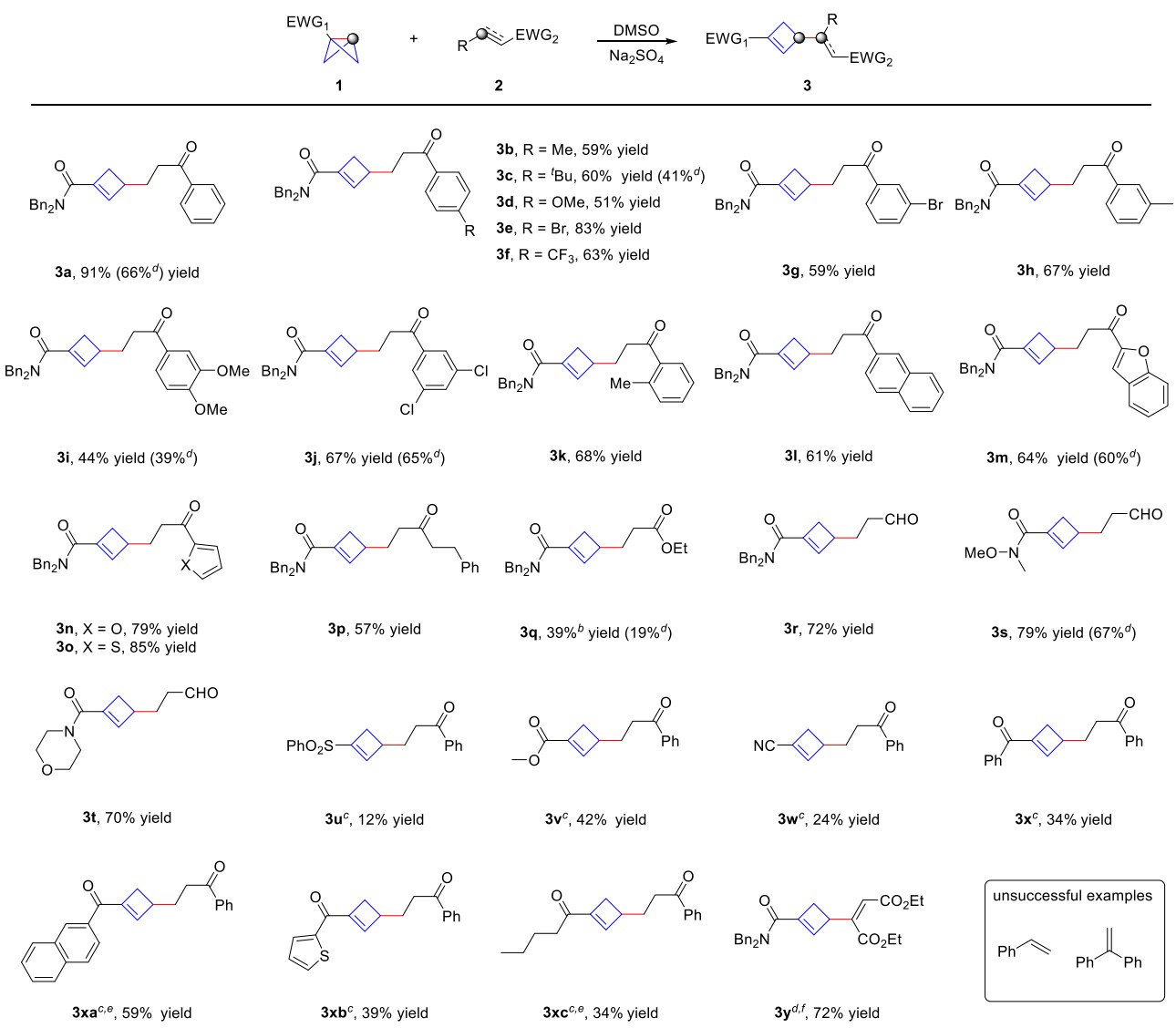

**Fig. 3 | Substrate scope of monosubstituted alkenes.**[a] **a** Reaction conditions A: BCB **1** (0.2 mmol), alkene **2** (0.1 mmol), Na$_2$SO$_4$ (0.17 mmol), 80 °C in DMSO (0.1 mL) under argon, 24 h, isolated yield. **b** 100 °C. **c** neat without any solvent, BCB **1** (0.1 mmol), alkene **2** (0.5 mmol), Na$_2$SO$_4$ (0.17 mmol). **d** Reaction conditions B: BCB **1** (0.1 mmol), alkene **2** (0.4 mmol), Na$_2$SO$_4$ (0.17 mmol), 80 °C in DMSO (0.1 mL). **e** MgSO$_4$ instead of Na$_2$SO$_4$. **f** 40 °C.

## Mechanistic studies

Then we conducted several experiments to probe the reaction mechanism[38–41]. When CF$_3$CD$_2$OD was used as solvent or 20.0 eq CD$_3$CO$_2$D was added in DMSO solvent, no deuterium incorporation was detected in the product **3a**, which excludes the carbanion species in the reaction (Fig. 7A). Besides, several radical-probe experiments were performed. The reaction of **1a** and **2a** was found to be essentially unaffected by the addition of 1, 1-diphenylethylene, BHT or TEMPO (2, 2, 6, 6-tetramethyl-1-piperidinyloxy) (Fig. 7B). The cyclopropyl-containing alkene **2y** was also successfully applied to deliver the 1, 4-conjugate addition products **14** and **15** without the cleavage of C−C bond in cyclopropyl group. Furthermore, the phenyl-substituted **2y′** was further used to probe the radical pathway, which also delivered cyclopropyl-containing products **16** and **17** (Fig. 7C). EPR experiments showed no radical signal observed for the formation of products **3a, 6a, 16** and **17** (for more details see Supplementary Figs. 41–44) (Fig. 7D)[60]. In addition, kinetic studies of the reaction of BCB **1a** and enone **2aa** exhibited first-order kinetics for both BCB **1a** and **2aa**, indicating that both BCB and alkene are involved in the turnover-limiting step (Fig. 7E). These kinetic results are consistent with the

density functional theory (DFT) calculations results in Fig. 8. The experimentally measured activation free energy from the Eyring equation ($\Delta G_{exp}^{\ddagger}$ = 28.1 kcal/mol, for more details see the Supplementary Figs. 45–49), these results agree with the DFT calculation of the energy barrier (**TS1** in Fig. 8, $\Delta G^{\ddagger}$ = 27.9 kcal/mol) (Fig. 7F)[61].

Theoretical calculations were performed to investigate the detailed mechanism of this umpolung addition (for more computational details, see the Supporting Information). As shown in Fig. 8, both the α site and β site of **1a** were considered to undergo an Alder-ene type reaction with alkene **2aa**, where the middle C−C σ bond cleavage in BCB **1a** was broken accompanying by a hydrogen transfer from four-membered ring to alkene. The computational results demonstrate that the energy barrier of Alder-ene type reaction on β site via transition state **TS1** ($\Delta G^{\ddagger}$ = 27.9 kcal/mol) is 6.9 kcal/mol lower than that on α site via transition state **TS1′** ($\Delta G^{\ddagger}$ = 34.8 kcal/mol). Therefore, **4a** is the main product in kinetically, which is consistent with the chemoselectivity in experiment. In addition, when the reaction temperature is raised from 80 °C to 130 °C, the four-membered ring opening of **4a** could irreversibly occur to generate the diene product **INT1** via pericyclic transition state **TS2** ($\Delta G^{\ddagger}$ = 32.9 kcal/mol). Isomerization of diene **INT1**

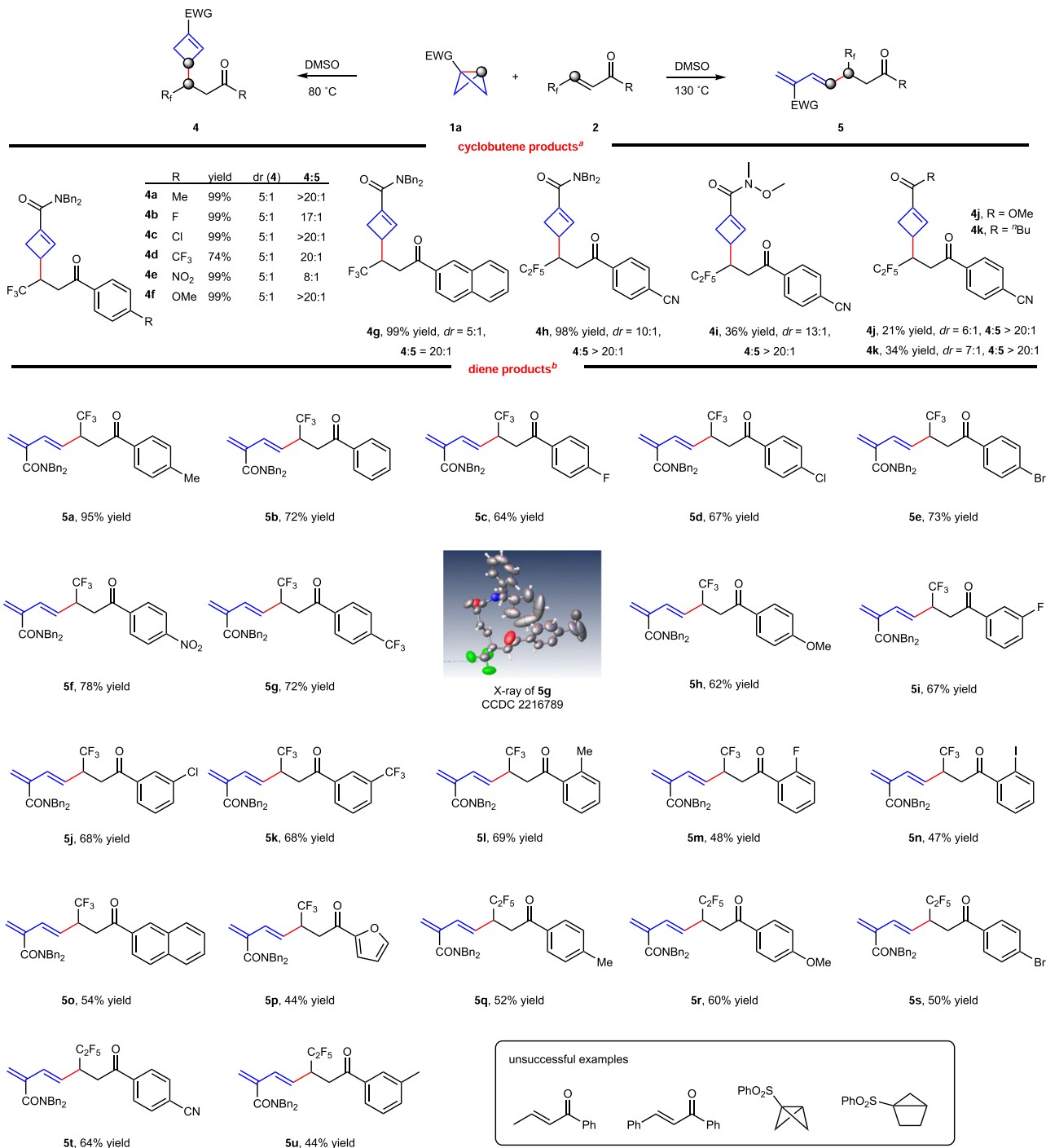

**Fig. 4 | Substrate Scope of 1, 2-disubstituted alkenes.** **a** Reaction conditions: BCB **1** (0.3 mmol), alkene **2** (0.1 mmol), Na₂SO₄ (0.17 mmol), 80 °C in DMSO (0.1 mL) under argon, 48 h, isolated yield. **b** Reaction conditions: BCB **1** (0.2 mmol), alkene **2** (0.1 mmol), Na₂SO₄ (0.17 mmol), 130 °C in DMSO (1.0 mL) under argon, 48 h, isolated yield.

would give the diene product **5a**. All the calculated results are consistent with the experimental results. When the BCB substrate has a phenyl group at the β site (**1a′**), the Alder-ene type reaction at the α site should be more energetically favorable to deliver the product **6a**, and the regioselectivity could be reversed by replacing the substrate **1a** with **1a′** having a phenyl substituent at the β site, probably due to the steric hindrance effect, which is also in agreement with the experimental result (for more details see the Supplementary Fig. 57). During our submission, Anderson's group also discovered the stereoselective Alder-ene reactions of bicyclo[1.1.0]butanes with strained alkenes and alkynes. The regioselectivity is the same as our product **6** in Fig. 5 and

their mechanism studies also support an Alder-ene process, not a radical pathway[62]. The concerted transition state of the C−C σ bond cleavage in BCBs together with hydrogen transfer contributes to the chemoselectivity.

In summary, we have developed a formal 1,4-conjugate addition of polarity-reversal C−C σ-bond to activated alkene under very simple reaction conditions without transition-metal catalysis, in which the strain-release-driven strategy is crucial for this neutral transformation. The umpolung reaction of C−C σ-bonds in bicyclo[1.1.0]butanes (BCBs) with various electrophilic alkenes, provides an atom-economic and straightforward approach for the synthesis of cyclobutenes and

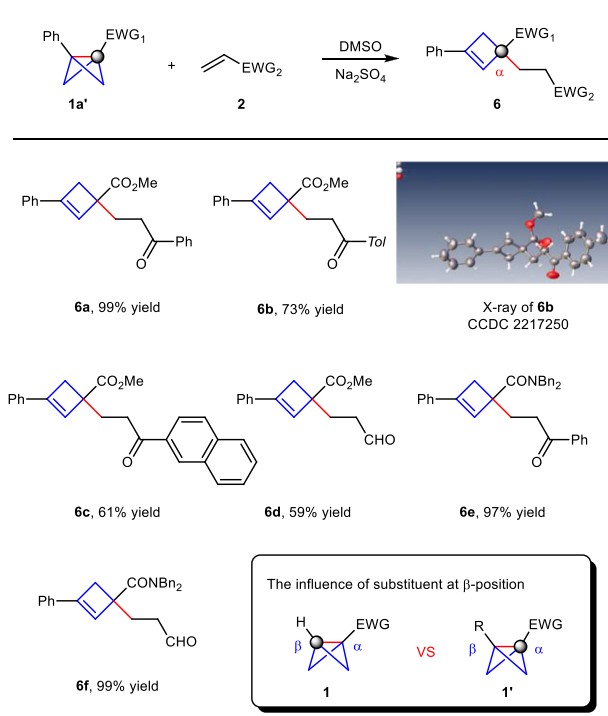

**Fig. 5 | The influence of the substituent at the β-position of bicyclo[1.1.0] butanes.[a]** **a** Reaction conditions: BCB **1a'** (0.2 mmol), alkene **2** (0.1 mmol), Na₂SO₄ (0.17 mmol), 80 °C in DMSO (0.1 mL) under argon, 24 h, isolated yield.

conjugated dienes with excellent selectivity. Mechanistic studies showed that a concerted Alder-ene type process would be the most likely pathway to give these formal conjugate addition products. We anticipate that this synthetic protocol will enhance chemists' tools for classic umpolung reactions. From a broader perspective, we envision that this umpolung reaction of strained C–C σ-bonds strategy will prove applicable in both modern synthetic chemistry and pharmaceutical research.

## Methods

### Umpolung reactions of polarized C–C σ-bonds with activated alkenes

**1a** (0.2 mmol), **2** (0.1 mmol), Na₂SO₄ (0.17 mmol) in DMSO (0.1 mL) were charged into a pressure tube under argon. The reaction tube was then sealed and placed into an oil bath at 80 °C. After being stirred for 24 h, the reaction vessel was removed from the oil bath and cooled to ambient temperature. To the reaction mixture, H₂O (4.0 mL) was added, and the mixture was extracted with Et₂O. The organic layer was concentrated and purified by silica gel chromatography (PE: EA = 5:1) to give the indicated product **3** or **6**. More details and characterization of the products are available in the Supplementary Information.

**1a** (0.3 mmol), **2** (0.1 mmol), Na₂SO₄ (0.17 mmol) in DMSO (0.1 mL) were charged into a pressure tube under argon. The reaction tube was then sealed and placed into an oil bath at 80 °C. After being stirred for 24 h, the reaction vessel was removed from the oil bath and cooled to ambient temperature. To the reaction mixture, H₂O (4.0 mL) was added, and the mixture was extracted with Et₂O. The organic layer was concentrated and purified by silica gel chromatography (PE: EA = 5:1) to give the indicated product **4**, More

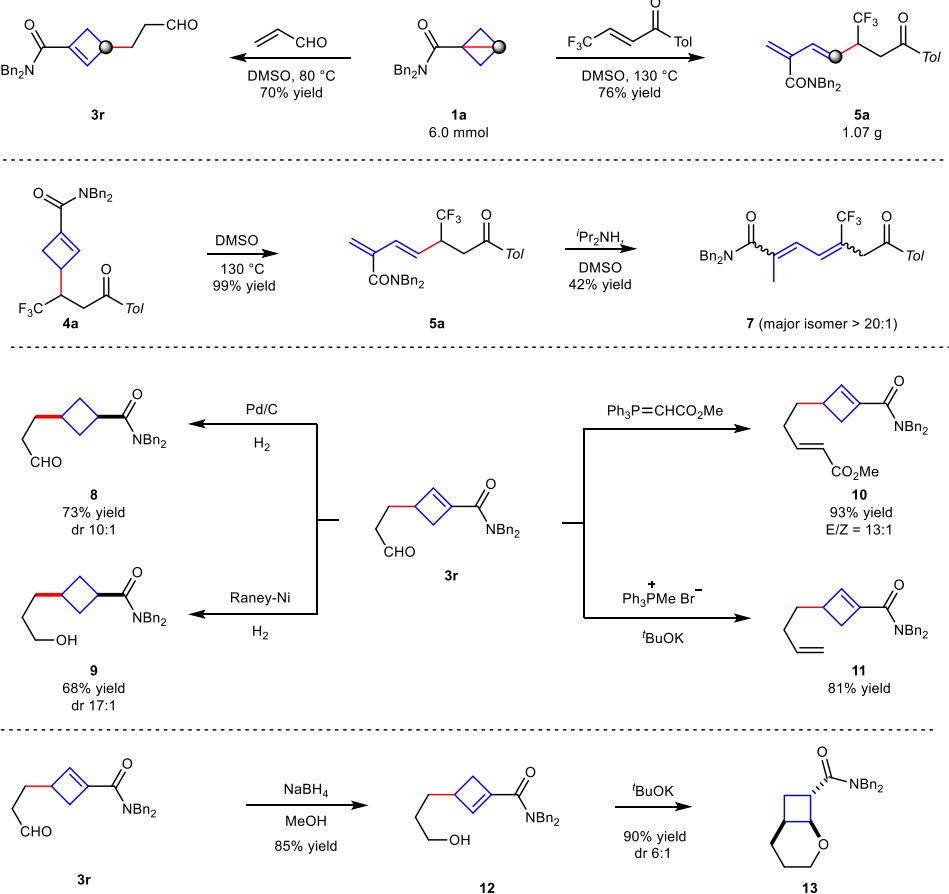

**Fig. 6 | Synthetic applications.** Scale-up synthesis and derivatization of **3r**.

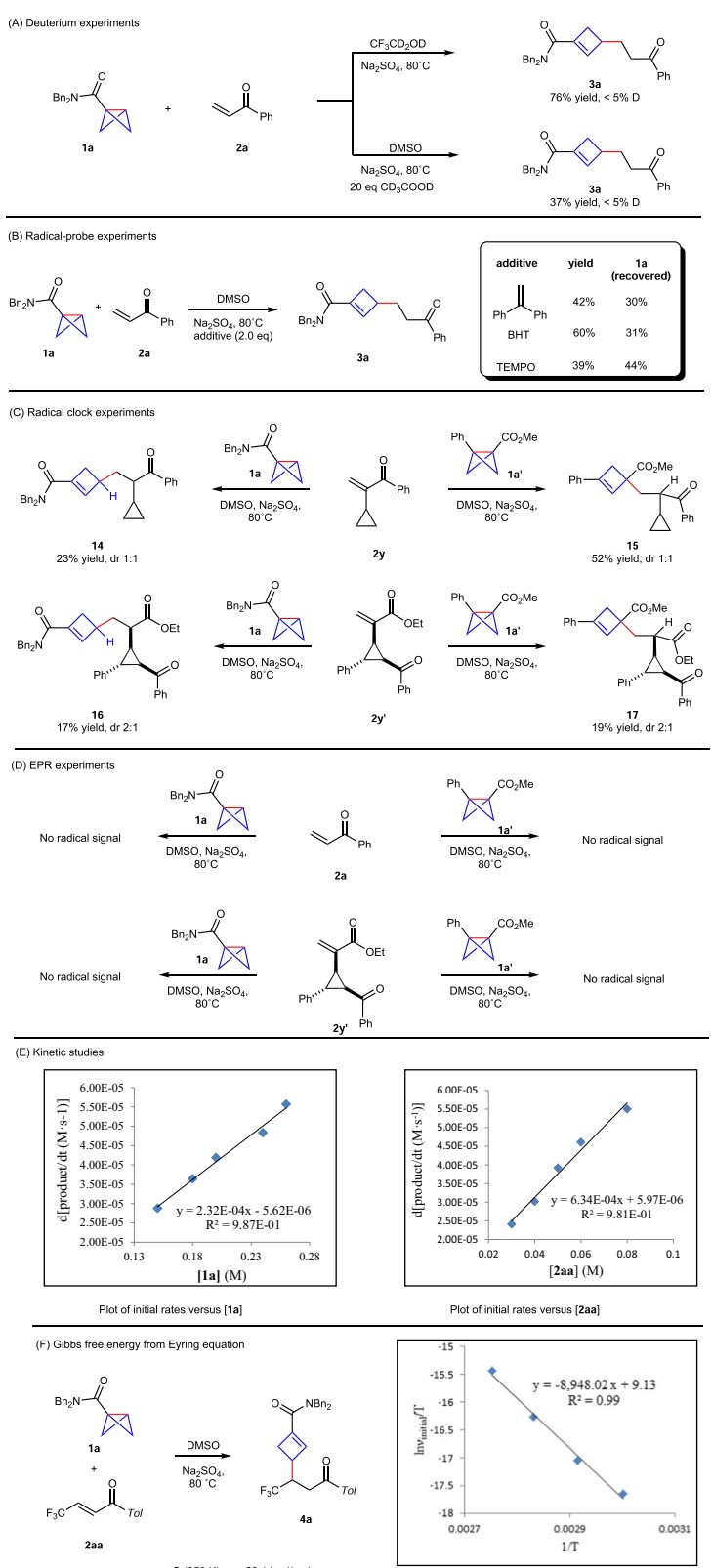

**Fig. 7 | Mechanistic studies. A** Deuterium experiments. **B** Radical-probe experiments. **C** Radical clock experiments. **D** EPR experiments. **E** Kinetic studies. **F** The experimentally measured Gibbs free energy.

details and characterization of the products are available in the Supplementary Information.

**1a** (0.2 mmol), **2** (0.1 mmol), Na$_2$SO$_4$ (0.17 mmol) in DMSO (1.0 mL) were charged into a pressure tube under argon. The reaction tube was then sealed and placed into an oil bath at 130 °C. After

being stirred for 48 h, the reaction vessel was removed from the oil bath and cooled to ambient temperature. To the reaction mixture, H$_2$O (4.0 mL) was added, and the mixture was extracted with Et$_2$O. The organic layer was concentrated and purified by silica gel chromatography (PE: EA = 5:1) to give the indicated product **5**. More

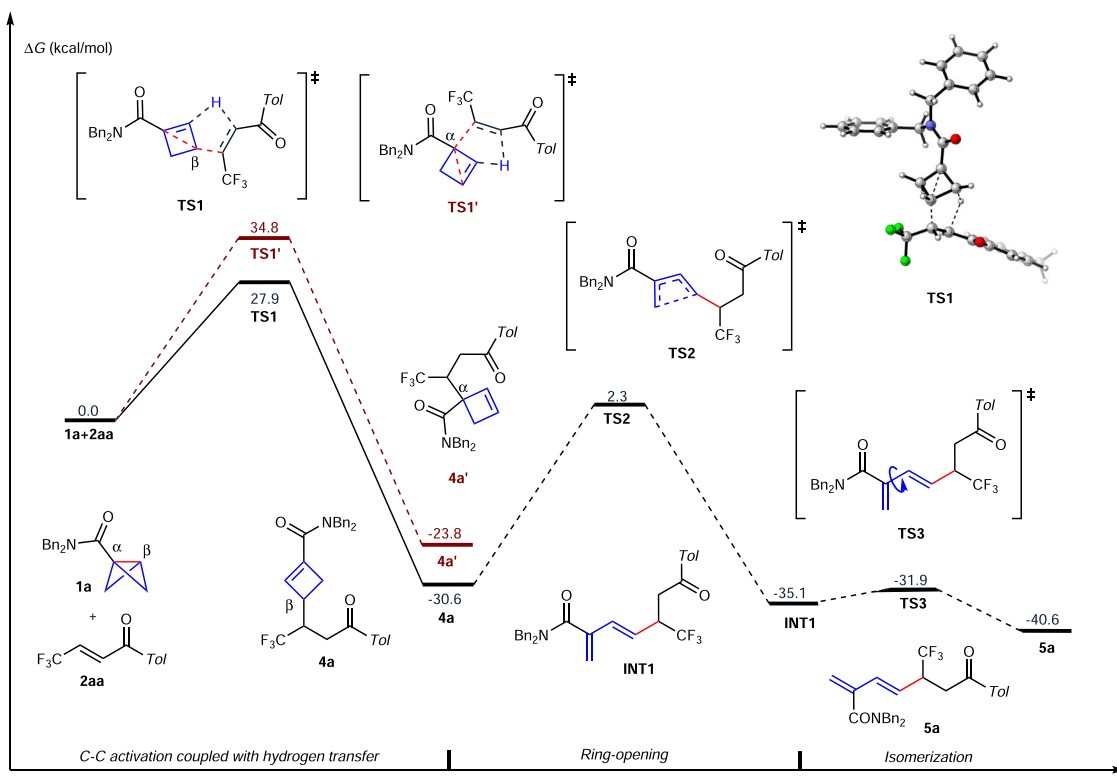

**Fig. 8 | Computational studies.** The corresponding energy profiles of the chemoselective pathways for the selected model reaction between BCB **1a** and β-trifluoromethyl enone **2aa**.

details and characterization of the products are available in Supplementary Information.

## Data availability

The authors declare that the data supporting the findings of this study are available within the paper and its Supplementary Information files, and are also available from the corresponding author. The nuclear magnetic resonance (NMR), experimental procedures and characterization for all products, mechanism studies are shown in the Supplementary Information files. The X-ray crystallographic coordinates for the structures reported in this article have been deposited at the Cambridge Crystallographic Data Centre (CCDC), under deposition numbers CCDC 2216789 (for **5g**) and CCDC 2217250 (for **6b**). These data can be obtained free of charge from the Cambridge Crystallographic Data Centre via www.ccdc.cam.ac.uk/structures.

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

## Acknowledgements

We thank Prof. Hongli Bao for the kind discussion of radical-probe experiments. This work is supported by the NSFC (Nos. 82130103 (J.C.), U1804283 (J.C.), 21801067 (D.B.)), the Central Plains Scholars and Scientists Studio Fund (2018002 (J.C.)), and the Project funded by the Natural Science Foundation of Henan (202300410225 (D.B.), 222102310562 (H.W.)) and Henan Postdoctoral Science Foundation (202103087 (H.W.)). We also thank the financial support from Henan Key Laboratory of Organic Functional Molecules and Drug Innovation.

## Author contributions
D.B. initiated the project, designed and directed the project, completed products characterizations and wrote the manuscript; X.G., W.X., R.C. did some experiments and some analysis of products; X.W., D.W. and Y.L. did DFT calculation; Y.L and J.C. supported the project and wrote the manuscript. J.C. also directed the project. X.G., X.W. and W.X. contributed equally.

## Competing interests
The authors declare no competing interests.
