## [Peer Review File · Nature Communications]

REVIEWER COMMENTS

Reviewer #1 (Remarks to the Author):

The manuscript submitted by Bai, Chang, and co-workers describes a metal-free formal 1,4-conjugate addition of polarity-reversal C–C σ -bonds of bicyclobutanes (BCBs) to activated alkenes. This results in the formation of functionalized cyclobutene products with yields that range from poor to moderate in most cases. Moreover, the scope of bicyclobutane is restricted to monosubstituted forms. The reaction yielded the expected product of the Alder-ene reaction when employing 1,3-disubstituted bicyclobutanes as the substrate. Considering that regular Alder-ene reactions of 1,3-disubstituted bicyclobutanes with alkenes have already been discovered (ref 43-44 and 48-49), and the umpolung Alder-ene reaction of BCBs have also been reported by Biju and Anderson (ref 62), this referee does not consider this paper to be at the level of a Nature Communications contribution due to a lack of conceptual novelty.

Reviewer #2 (Remarks to the Author):

This manuscript described a metal-free umpolung reaction enabled by strain-release of bicyclo[1.1.0]butanes. This reaction represented a polarity reversal of C–C σ -bond under simple reaction conditions and provide a new way to access cyclobutene derivatives. The mechanism studies provide supporting evidence that the reaction underwent an Alder-ene type reaction with central σ -bond cleavage and proton transfer. Overall, I am very positive, but have a few concerns. One is that the scope of BCB is limited to those bearing an amide functionality. The second concern is on the authors' interpretation with respect to the formation of diene product. The ring opening of cyclobutene to diene, an electrocyclic ring opening reaction, is a well-known pericyclic reaction that was often taught in graduate level courses. It is almost certain that 5 is not formed directly from 2, instead 2 first was converted into 4 then 5. At 130 C, the process of 2 to 4 to 5 became a one-pot process. The authors should investigate that if 4 converts to 5 at 130 Celsius. The third one is that I do not know why figure 5 is in the manuscript, which showed that BCBs with *o*-aryl substituents the reaction went back to normal reactivity pattern. If the authors could address the three concerns, I would consider a recommendation for publication of this manuscript in Nature Communication.

There are many typos in the manuscripts:

Page 2 line 48 and 49: "we questioned whether the strain-release-driven enable the polarized C–C σ -bonds react with alkene umpolung." is not expressed exactly.

Page 4, line 71, "increase" should be changed to "increasing".

Page 9, line 138; Page 12, 174; Page13, Figure 8: “isomeration” should be revised to “isomerization”.

Page 9, line 139: “was successful hydrogen” should be changed to “was successfully hydrogenated”.

Page 9, line 142: “successful” should be its adverb “successfully”.

Page 9, line 143: “reductive” should be “reduction”.

Page 13, line 180: “submittion” should be “submission”.

Page 14, line 202 and 208: “exacted” should be corrected to “extracted”.

Some suggestions:

In SI, following the procedure A, the reaction scale is 0.1 mmol and some cases were set up in 0.2 mmol scale (i.e. 3d-3k). It is necessary to make a clear note of the exact scale in specific reaction.

Reviewer #3 (Remarks to the Author):

Chang et al. reported a formal 1, 4-conjugate addition of polarity-reversal C–C σ -bond to activated alkene without transition-metal catalysis. This work extends the umpolung reaction to C–C σ -bonds, which has certain value and may become a new synthetic tool for chemists. The authors used a combination of experimental and theoretical methods. The theoretical calculation method and the choice of the basis group are appropriate, and the calculated total potential barrier is also reasonable under the current reaction conditions. In the theoretical calculations section, I don't find any points that should be corrected. So, I recommend its publication in Nature Communications.

REVIEWER COMMENTS

Reviewer #1 (Remarks to the Author):

The manuscript submitted by Bai, Chang, and co-workers describes a metal-free formal 1,4-conjugate addition of polarity-reversal C–C σ -bonds of bicyclobutanes (BCBs) to activated alkenes. This results in the formation of functionalized cyclobutene products with yields that range from poor to moderate in most cases. Moreover, the scope of bicyclobutane is restricted to monosubstituted forms. The reaction yielded the expected product of the Alder-ene reaction when employing 1,3-disubstituted bicyclobutanes as the substrate. Considering that regular Alder-ene reactions of 1,3-disubstituted bicyclobutanes with alkenes have already been discovered (ref 43-44 and 48-49), and the umpolung Alder-ene reaction of BCBs have also been reported by Biju and Anderson (ref 62), this referee does not consider this paper to be at the level of a Nature Communications contribution due to a lack of conceptual novelty.

Reviewer #2 (Remarks to the Author):

This manuscript described a metal-free umpolung reaction enabled by strain-release of bicyclo[1.1.0]butanes. This reaction represented a polarity reversal of C-C σ -bond under simple reaction conditions and provide a new way to access cyclobutene derivatives. The mechanism studies provide supporting evidence that the reaction underwent an Alder-ene type reaction with central σ -bond cleavage and proton transfer. Overall, I am very positive, but have a few concerns. One is that the scope of BCB is limited to those bearing an amide functionality. The second concern is on the authors' interpretation with respect to the formation of diene product. The ring opening of cyclobutene to diene, an electrocyclic ring opening reaction, is a well-known pericyclic reaction that was often taught in graduate level courses. It is almost certain that 5 is not formed directly from 2, instead 2 first was converted into 4 then 5. At 130 C, the process of 2 to 4 to 5 became a one-pot process. The authors should investigate that if 4 converts to 5 at 130 Celsius. The third one is that I do not know why figure 5 is in the manuscript, which showed that BCBs with *o*-aryl substituents the reaction went back to normal reactivity pattern. If the authors could address the three concerns, I would consider a recommendation for publication of this manuscript in Nature Communication.

There are many typos in the manuscripts:

Page 2 line 48 and 49: “we questioned whether the strain-release-driven enable the polarized C-C σ -bonds react with alkene umpolung.” is not expressed exactly.

Page 4, line 71, “increase” should be changed to “increasing”.

Page 9, line 138; Page 12, 174; Page13, Figure 8: “isomeration” should be revised to “isomerization”.

Page 9, line 139: “was successful hydrogen” should be changed to "was successfully hydrogenated”.

Page 9, line 142: “successful” should be its adverb “successfully”.

Page 9, line 143: “reductive” should be “reduction”.

Page 13, line 180: “submission” should be “submission”.

Page 14, line 202 and 208: “exacted” should be corrected to “extracted”.

Some suggestions:

In SI, following the procedure A, the reaction scale is 0.1 mmol and some cases were set up in 0.2 mmol scale (i.e. 3d-3k). It is necessary to make a clear note of the exact scale in specific reaction.

Reviewer #3 (Remarks to the Author):

Chang et al. reported a formal 1, 4-conjugate addition of polarity-reversal C–C σ -bond to activated alkene without transition-metal catalysis. This work extends the umpolung reaction to C–C σ -bonds, which has certain value and may become a new synthetic tool for chemists. The authors used a combination of experimental and theoretical methods. The theoretical calculation method and the choice of the basis group are appropriate, and the calculated total potential barrier is also reasonable under the current reaction conditions. In the theoretical calculations section, I don't find any points that should be corrected. So, I recommend its publication in Nature Communications.

To the comments of Reviewer 1:

Comments: The manuscript submitted by Bai, Chang, and co-workers describes a metal-free formal 1,4-conjugate addition of polarity-reversal C–C σ -bonds of bicyclobutanes (BCBs) to activated alkenes. This results in the formation of functionalized cyclobutene products with yields that range from poor to moderate in most cases. Moreover, the scope of bicyclobutane is restricted to monosubstituted forms. The reaction yielded the expected product of the Alder-ene reaction when employing 1,3-disubstituted bicyclobutanes as the substrate. Considering that regular Alder-ene reactions of 1, 3-disubstituted bicyclobutanes with alkenes have already been discovered (ref 43-44 and 48-49), and the umpolung Alder-ene reaction of BCBs have also been reported by Biju and Anderson (ref 62), this referee does not consider this paper to be at the level of a Nature Communications contribution due to a lack of conceptual novelty.

Reply: We agree with this reviewer that some regular Alder-ene reactions of BCBs have been discovered (ref 43-44 and 48-49), but this reviewer might not read Ref. 62 carefully and made an important mistake. The system developed by Biju and Anderson (Ref 62) is also regular Alder-ene reaction, but not umpolung system. The regioselectivity from Biju and Anderson's work (Ref 62) is contrary to our main umpolung systems (**Fig. 3** and **Fig. 4**), but same with our control experiments in **Fig. 5**. Up to data, our system is also the first umpolung Alder-ene reaction of BCBs. Accordingly, we would like to argue that this reviewer's decision is somewhat imbalanced.

The main purpose of this manuscript is the Umpolung reactivity of strained C–C σ -Bonds in BCBs without any transition-metal catalysts. To the best of our knowledge, only two examples of polarity-reversal strategy with BCBs by transition-metal catalyst have been reported during our manuscript submission, and both of them proceeded through radical pathway (Co-catalyst: *J. Am. Chem. Soc.* **2020**, *142*, 5355–5361.; and SmI₂ catalyst: *Nat. Chem.* **2023**, *15*, 535–541.). Our studies realized the umpolung C–C σ -bonds of BCBs through Alder-ene process, the regioselectivity is opposite to all of previous Alder-ene reactions (Ref 43-44, 48-49 and Ref 62). Broad functional groups were tolerated and most cases gave synthetically useful yields (67 examples overall, more than 50 examples gave 50-99% yields). Moreover, we also demonstrated several derivative reactions and detailed mechanism studies. This study developed the primary transition-metal free umpolung reaction of C–C σ -bonds and would provide a new strategy for the transformation of BCBs.

As Ref 62 has been published online on *JACS* very recently (<https://doi.org/10.1021/jacs.3c13080>), so we changed Ref 62 from *ChemRxiv* to *JACS*.

To the comments of Reviewer 2:

Comments: This manuscript described a metal-free umpolung reaction enabled by strain-release of bicyclo[1.1.0]butanes. This reaction represented a polarity reversal of C–C σ -bond under simple reaction conditions and provide a new way to access cyclobutene derivatives. The mechanism studies provide supporting evidence that the reaction underwent an Alder-ene type reaction with central σ -bond cleavage and proton transfer. Overall, I am very positive, but have a few concerns.

Reply: We sincerely thank the reviewer's very positive comments.

Comment 1: One is that the scope of BCB is limited to those bearing an amide functionality.

Reply: Thank you very much for the valuable comment. To better understand the substrate scope of this system, several experiments were performed, as shown below (also compiled in **Fig. 3** and **Fig. 4** of the revised manuscript):

For monosubstituted alkenes:

Beside dibenzyl amide group, other amides including Weinreb amide and morpholine amide were tolerated (**3s**, 79% yield; **3t**, 70% yield). Various electron-withdrawing groups including sulfone, ester, nitrile, *aryl*- ketone and *alkyl*- ketones are also successfully converted to the corresponding products just with modified reaction conditions (**3u**, 12% yield; **3v**, 42% yield; **3w**, 24% yield; **3x**, 34% yield; **3xa**, 59% yield; **3xb**, 39% yield and **3xc**, 34% yield). These results have been added in **Fig. 3** of the revised manuscript.

[a] Reaction conditions A: **1** (0.2 mmol), **2** (0.1 mmol), Na₂SO₄ (0.17 mmol), 80 °C in DMSO (0.1 mL) under argon, 24 h, isolated yield. [b] Reaction conditions B: neat without any solvent, **1** (0.1 mmol), **2** (0.5 mmol), Na₂SO₄ (0.17 mmol). [c] MgSO₄ instead of Na₂SO₄.

First, we briefly investigated other amide groups and found that Weinreb amide and morpholine amide worked smoothly to deliver the corresponding products (**3s**, 79% yield; **3t**, 70% yield). Then, other electron-withdrawing groups instead of amides such as sulfone, ester, nitrile and ketone were tested, but all of these systems gave trace amount of desired products under the previous optimized reaction conditions (Reaction conditions A). So, we investigated the reaction conditions of other electron-withdrawing groups attached BCBs. As shown below, the reaction of **1u** with **2a** could give product **3u** in 12% yield under neat conditions (Reaction conditions B). We corrected the yield of **3u** and to avoid such a mistake, we repeated all of reactions in **Fig. 3** and **Fig. 4** and double-checked the yield of these reactions.

For sulfone attached BCB:

Entry	Variation from the "standard" condition ^a	3u , yield (%)
1	none	5
2	1 eq Na ₂ SO ₄ instead of 1.6 eq Na ₂ SO ₄	2
3	MgSO ₄ instead of Na ₂ SO ₄	6
4	toluene instead of DMSO	8
5	dioxane instead of DMSO	7
6	DMF instead of DMSO	NR
7	DMA instead of DMSO	2
8	DCE instead of DMSO	4
9	MeOH instead of DMSO	2
10	CH ₃ CN instead of DMSO	4
11	50 °C instead of 80 °C	NR
12	100 °C instead of 80 °C	8
13	0.05 mL DMSO instead of 0.1 mL DMSO	7
14 ^b	5 eq 1u used	12

[a] Reaction conditions A: **2a** (0.1 mmol), **1u** (0.2 mmol), Na₂SO₄ (0.17 mmol), 80 °C in DMSO (0.1 mL) under argon, 24 h, isolated yield. [b] Reaction conditions B: **2a** (0.5 mmol), **1u** (0.1 mmol).

Other functional groups including ester, nitrile and aryl ketone were investigated. These substrates could give umpolung Alder-ene products under neat conditions (**3v**, 42% yield; **3w**, 24% yield; **3x**, 34% yield).

To better define the substrates scope, we finally investigated other functionalized groups such as different *aryl*-substituted ketones and *alkyl*-substituted ketone, which worked smoothly to deliver

desired products just with modified reaction conditions (**3xa**, 59% yield; **3xb**, 39% yield and **3xc**, 34% yield).

For naphthyl ketone attached BCB:

Entry	Variation from the "standard" condition ^a	3xa , yield (%)
1	none	19
2	1 eq Na_2SO_4 instead of 1.7 eq Na_2SO_4	18
3	MgSO_4 instead of Na_2SO_4	18
4	tol instead of DMSO	18
5	dioxane instead of DMSO	16
6	DMA instead of DMSO	12
7	DCE instead of DMSO	20
8	MeOH instead of DMSO	NR
9	CH_3CN instead of DMSO	26
10	$50\text{ }^\circ\text{C}$ instead of $80\text{ }^\circ\text{C}$	11
11	$100\text{ }^\circ\text{C}$ instead of $80\text{ }^\circ\text{C}$	NR
12 ^b	5 eq 2a used	14
13 ^b	$50\text{ }^\circ\text{C}$ instead of $80\text{ }^\circ\text{C}$	52
14 ^{b,c}	no Na_2SO_4	50
15 ^{b,c}	MgSO_4 instead of Na_2SO_4	59

[a] Reaction conditions A: **2a** (0.1 mmol), **1xa** (0.2 mmol), Na_2SO_4 (0.17 mmol), $80\text{ }^\circ\text{C}$ in DMSO (0.1 mL) under argon, 24h, isolated yield. [b] Reaction conditions B: **2a** (0.5 mmol), **1xa** (0.1 mmol), Na_2SO_4 (0.17 mmol), neat without any solvent, $80\text{ }^\circ\text{C}$ under argon, 48 h, isolated yield. [c] $50\text{ }^\circ\text{C}$.

For thienyl ketone attached BCB:

Entry	Variation from the "standard" condition ^a	3xb , yield (%)
1	none	39
2	no Na_2SO_4	26
3	MgSO_4 instead of Na_2SO_4	21
4 ^b	2 eq 1xb used	NR
5 ^b	$80\text{ }^\circ\text{C}$ instead of $50\text{ }^\circ\text{C}$	NR

[a] Reaction conditions A: **2a** (0.5 mmol), **1xb** (0.1 mmol), Na_2SO_4 (0.17 mmol), neat without any solvent, $50\text{ }^\circ\text{C}$ under argon, 48 h, isolated yield. [b] Reaction conditions B: **2a** (0.1 mmol), **1xb** (0.2 mmol), MgSO_4 (0.2 mmol), $50\text{ }^\circ\text{C}$ in DMSO (0.1 mL) under argon, 24h, isolated yield.

For alkyl ketone attached BCB:

Entry	Variation from the "standard" condition ^a	3xc, yield (%)
1	none	34
2	no MgSO ₄	15
3	Na ₂ SO ₄ instead of MgSO ₄	12
4	30 °C instead of 50 °C	26
5	80 °C instead of 50 °C	18
6	1 eq MgSO ₄ instead of 2 eq MgSO ₄	32
7	3 eq MgSO ₄ instead of 2 eq MgSO ₄	34
8 ^b	2 eq 1xc	5
9 ^b	80 °C instead of 50 °C	10

[a] Reaction conditions A: **2a** (0.5 mmol), **1xc** (0.1 mmol), MgSO₄ (0.2 mmol), neat without any solvent, 50 °C under argon, 48 h, isolated yield. [b] Reaction conditions B: **2a** (0.1 mmol), **1xc** (0.2 mmol), Na₂SO₄ (0.17 mmol), 50 °C in DMSO (0.1 mL) under argon, 24h, isolated yield.

For challenged 1, 2-disubstituted alkenes:

We investigated other electron-withdrawing groups substituted BCBs for more challenged 1, 2-disubstituted alkenes. Only trace amount of desired product obtained for sulfone or nitrile attached BCBs. We finally realized the systems of Weinreb amide, ester and alkyl ketone attached BCBs, which gave umpolung addition product **4i** in 36% yield (*dr* 13: 1, **4:5** > 20:1), **4j** in 21% yield (*dr* 6: 1, **4:5** > 20:1) and **4k** in 34% yield (*dr* 7: 1, **4:5** > 20:1), respectively. These results have been added in **Fig. 4** of the revised manuscript. For these functionalized BCBs, only trace amount of corresponding diene product **5** was observed.

In the revised manuscript, we have added the result of **3xa**, **3xb** and **3xc** in Figure 3. We also added the sentence “Aryl ketones and alkyl ketone attached BCBs also worked smoothly to give umpolung Alder-ene products (**3x**, 34% yield, **3xa**, 59% yield; **3xb**, 39% yield and **3xc**, 34% yield) (for details to see the Supplementary Table S2 and Table S3).” in paragraph 5.

We have added the result of **4i**, **4j** and **4k** in Figure 4. We also added the sentence “Other electron-withdrawing groups such as Weinreb amide (**4i**, 36% yield) , ester (**4j**, 21% yield) and alkyl ketone (**4j**, 34% yield) did lead to umpolung addition product formation, albeit with low yields.” in paragraph 6.

In the Supporting information, we have added ¹H NMR, ¹³C NMR and HRMS of products **3xa**, **3xb**, **3xc**, **4i** and **4j**. We also added the optimization of reaction conditions of **1u** as Supplementary Table S2 and **1xa** as Supplementary Table S3.

Comment 2: The second concern is on the authors’ interpretation with respect to the formation of diene product. The ring opening of cyclobutene to diene, an electrocyclic ring opening reaction, is a well-known pericyclic reaction that was often taught in graduate level courses. It is almost certain that **5** is not formed directly from **2**, instead **2** first was converted into **4** then **5**. At 130 C, the process of **2** to **4** to **5** became a one-pot process. The authors should investigate that if **4** converts to **5** at 130 Celsius.

Reply: Thank you very much for the great suggestion. It is true that diene product **5** is not formed directly from the reaction of **2** and **1**. The reaction of **2** and **1** would give product **4** firstly and then converted to diene product **5** through electrocyclic ring-opening process.

The diene product **5a** could be obtained quantitatively (99% yield) from **4a** through electrocyclic ring-opening process at 130°C under our reaction conditions. This result has been added in Fig. 6.

In the revised manuscript, we have added the sentence “Primary study indicated diene product **5a** was formed from **4a** through electrocyclic ring-opening process (Fig. 6)” in paragraph 6 to interpret the formation of diene product.

Comment 3: The third one is that I do not know why figure 5 is in the manuscript, which showed that BCBs with β -aryl substituents the reaction went back to normal reactivity pattern.

Reply: Thank you very much for the comment. These control experiments in Figure 5 showed that the reaction would go back to normal regioselectivity when BCBs with substituent at β -position. These results in Figure 5 would demonstrate the effect of β site substitution on BCBs in this polarity-reversed systems. Moreover, DFT calculations of β -aryl substituted BCBs also agree with the alder-ene pathway (Supplementary Figure S6).

Comment 4: There are many typos in the manuscripts:

1. Page 2 line 48 and 49: “we questioned whether the strain-release-driven enable the polarized C-C σ -bonds react with alkene umpolung.” is not expressed exactly.
2. Page 4, line 71, “increase” should be changed to “increasing”.
3. Page 9, line 138; Page 12, 174; Page13, Figure 8: “isomeration” should be revised to “isomerization”.
4. Page 9, line 139: “was successful hydrogen” should be changed to “was successfully hydrogenated”.
5. Page 9, line 142: “successful” should be its adverb “successfully”.
6. Page 9, line 143: “reductive” should be “reduction”.
7. Page 13, line 180: “submittion” should be “submission”.
8. Page 14, line 202 and 208: “exacted” should be corrected to “extracted”.

Some suggestions:

9. In SI, following the procedure A, the reaction scale is 0.1 mmol and some cases were set up in 0.2 mmol scale (i.e. 3d-3k). It is necessary to make a clear note of the exact scale in specific reaction.

Reply: We sincerely thank you for these corrections and suggestions.

1. We have changed the sentence “we questioned whether the strain-release-driven enable the polarized C-C σ -bonds react with alkene umpolung.” to “we questioned whether the strain-release-driven strategy could enable umpolung reaction of the polarized C-C σ -bond with alkene.”
2. We have changed “increase” to “increasing”.

3. We have corrected “isomeration” to “isomerization”.
4. We have changed “was successful hydrogen” to “was successfully hydrogenated”
5. We have changed “successful” to “successfully”.
6. We have changed “reductive” to “reduction”.
7. We have changed “submission” to “submittion”.
8. We have changed “exacted” to “extracted”.
9. In SI, we have added all of the reaction scale in SI.

To the comments of Reviewer 3:

Comments: Chang et al. reported a formal 1, 4-conjugate addition of polarity-reversal C–C σ -bond to activated alkene without transition-metal catalysis. This work extends the umpolung reaction to C–C σ -bonds, which has certain value and may become a new synthetic tool for chemists. The authors used a combination of experimental and theoretical methods. The theoretical calculation method and the choice of the basis group are appropriate, and the calculated total potential barrier is also reasonable under the current reaction conditions. In the theoretical calculations section, I don't find any points that should be corrected. So, I recommend its publication in Nature Communications.

Reply: We sincerely thank the reviewer for agreeing to publish our manuscript.

Other changes:

Other typos were corrected in the revised manuscript and Supporting Information.

Attached please find our revised manuscript and Supporting information. We are pleased to answer any further questions.

Sincerely yours,

Dachang Bai

REVIEWERS' COMMENTS

Reviewer #2 (Remarks to the Author):

The authors have addressed this reviewer's concerns towards the previous manuscript. This revised manuscript is suitable for publication on Nature Communications.

However, there are still minor errors in this manuscript. Please check it carefully. In some cases, the word "umpolung" is misused, for example P14, line 198; In P14, line 187, "Bicyclo[1.1.0]butanes" should be changed to "bicyclo[1.1.0]butanes".